# Activities of Chromatin Remodeling Factors and Histone Chaperones and Their Effects in Root Apical Meristem Development

**DOI:** 10.3390/ijms21030771

**Published:** 2020-01-24

**Authors:** Huijia Kang, Di Wu, Tianyi Fan, Yan Zhu

**Affiliations:** State Key Laboratory of Genetic Engineering, Collaborative Innovation Center for Genetics and Development, International Associated Laboratory of CNRS-Fudan-HUNAU on Plant Epigenome Research, Department of Biochemistry, Institute of Plant Biology, School of Life Sciences, Fudan University, Shanghai 200438, China; 18110700079@fudan.edu.cn (H.K.); 18210700071@fudan.edu.cn (D.W.); fantianyi1998@163.com (T.F.)

**Keywords:** chromatin remodeling factor, histone chaperone, root apical meristem, chromatin stability, gene regulation

## Abstract

Eukaryotic genes are packaged into dynamic but stable chromatin structures to deal with transcriptional reprogramming and inheritance during development. Chromatin remodeling factors and histone chaperones are epigenetic factors that target nucleosomes and/or histones to establish and maintain proper chromatin structures during critical physiological processes such as DNA replication and transcriptional modulation. Root apical meristems are vital for plant root development. Regarding the well-characterized transcription factors involved in stem cell proliferation and differentiation, there is increasing evidence of the functional implications of epigenetic regulation in root apical meristem development. In this review, we focus on the activities of chromatin remodeling factors and histone chaperones in the root apical meristems of the model plant species Arabidopsis and rice.

## 1. Introduction

Epigenetics refers to the study of heritable changes in gene expression that do not involve alterations to the nucleotide sequence. During plant growth and development, the chromatin structure in the nuclei must be precisely modulated in coordination with the transcriptional reprogramming associated with cell proliferation and cell fate determination within meristem regions. Nucleosomes, which are the basic structural and functional units of chromatin, comprise approximately 146 bp of DNA wrapped around a globular histone octamer, each containing two molecules of H2A, H2B, H3, and H4 [1]. Nucleosomes are connected by a 20–200 bp DNA linker and form repeating structures that appear as “beads-on-a-string” in electron microscopy images. The nucleosome array is further compacted into a more condensed and highly heterogeneous structure, and the DNA dynamics are generally correlated with local chromatin accessibility. De-condensed or open chromatin regions are often associated with active transcription, whereas genes in a condensed chromatin structure tend to be transcriptionally inert [2,3]. Nucleosome-depleted regions are usually detected in the promoters of highly transcribed genes rather than in the promoters of non-expressed genes or genes with low expression levels. Plants have a highly dynamic chromatin organization, and less than 20% of the Arabidopsis and rice genomes contain regularly positioned nucleosomes [4,5]. The strong electrostatic interaction between DNA and histones impedes the efficient remodeling of nucleosomes at physiological ionic strength. Chromatin remodeling factors and histone chaperones, both of which belong to multi-protein families, target nucleosomes and/or histones as substrates and are considered to have the intrinsic ability to re-organize chromatin structures alone or as part of a protein complex.

## 2. Plant Chromatin Remodeling Factors

Chromatin remodeling factors and their associated complexes can use the energy generated by ATP hydrolysis to regulate histone–DNA interactions and alter nucleosome composition, position, and occupancy, thereby altering DNA accessibility and functionally remodeling local or global chromatin structures [6,7,8]. In animals and plants, the activities of chromatin remodeling factors influence the maintenance and differentiation of stem cell fates. All chromatin remodeling factors in complete eukaryotic genomes contain a common Snf2 ATPase domain. This domain includes several conserved helicase-related sequence motifs (motifs I–VI), with different sequence lengths between motif III and motif IV, forming a “split” ATPase domain as a highly conserved bipartite combination of SNF2_N and Helicase_C. The N-terminal SNF2_N is the helicase ATP-binding domain, which is separated from the C-terminal Helicase_C by an “insertion region” [9]. On the basis of the Snf2 domain characteristics, these chromatin remodeling factors were briefly classified into six groups. Each group can be further subdivided into several subfamilies that exhibit specific properties [9]. The plant chromatin remodeling factors in different subfamilies differ regarding the Snf2 domain motif arrangement and the existence of other domains, such as the PHD (plant homeodomain) domain, chromodomain, bromodomain, SANT, and RING domain [10,11]. In this review, we focus on the subfamilies with reported functional roles in plant root meristems, including the SWI/SNF (switch/sucrose non-fermentable), ISWI (imitation switch), CHD (chromodomain helicase DNA-binding), and INO80 (inositol auxotrophy 80) subfamilies.

### 2.1. SWI/SNF Subfamily

The genome of the model plant Arabidopsis encodes four SWI/SNF subfamily remodelers: BRM (BRAHMA), SYD (SPLAYED), and two structurally- and functionally-related proteins, MINU1/CHR12 and MINU2/CHR23 [10,12]. In rice, a single gene, *CHR719*, corresponds to the Arabidopsis *CHR12* and *CHR23* genes [11], implying that these two Arabidopsis genes may have originated from a relatively recent gene duplication event.

The SWI/SNF ATPases have common domains in their N-terminus, including the QLQ (Gln-Leu-Gln) domain and the HSA (helicase-SANT-associated) domain, both of which are involved in the interaction between SWI/SNF ATPases and other proteins [13]. A previous study revealed that SWI/SNF ATPase-dependent transcriptional activation requires protein–protein interactions involving the HSA domain and nuclear ARPs (actin-related proteins) in yeast [14]. Interestingly, only the Arabidopsis BRM has an additional C-terminal bromodomain, which functions as a reader of acetylated histones. The bromodomain, together with three DNA-binding regions at the C-terminus, is thought to constitute a nucleosome-binding module that mediates the BRM interaction with its substrate [15]. In contrast, SYD and two MINU chromatin remodelers in yeast and human cells anchor histones via SnAC (Snf2 ATP coupling) domain [16], which has not been verified in plants.

Plant genomes also encode protein homologs of multiple required subunits of metazoan SWI/SNF complexes, including SWI3 homologs (SWI3A–D), SWP73-type proteins (SWP73A and SWP73B), ARPs (ARP4 and 7), and the SNF5-type protein BSH (BUSHY), as well as several optional proteins such as the OSA-type and POLYBROMO-type proteins [17]. Gel filtration chromatographic analyses proved that the Arabidopsis SWI/SNF ATPase BRM is a large protein complex (1–2 MDa) [18]. A recent study confirmed that BRM is post-translationally modified by the SUMO ligase AtMMS21 (methyl methane sulfonate sensitivity 21), which can stabilize BRM via SUMOylation. A depletion of functional AtMMS21 decreases the abundance of BRM proteins [19].

Many SWI/SNF chromatin remodeling factors can disassemble or slide nucleosomes in vitro [6]. In yeast, SWI/SNF chromatin remodeling factors contribute to the transcription by RNA polymerase II by removing histones, which is necessary for the transcription of multiple genes [20]. However, other yeast and human studies have indicated that SWI/SNF chromatin remodeling factor activities may also suppress the transcription by RNA polymerase II [21,22]. In Arabidopsis, BRM interacts with the promoter and terminator regions of nearly 1000 genes, and can activate and repress the transcription of the target genes [23], suggesting that BRM effects on transcription may be modulated by other subunits in the same protein complexes, by protein complex-interacting proteins, or via diverse nucleosome targets in specific chromatin regions. The replacement of the canonical histone H2A in nucleosomes with the histone variant H2A.Z affects nucleosome stability and has been implicated in a wide range of transcriptional regulatory activities in plants [24]. There is evidence that the Arabidopsis chromatin remodeling factors BRAHMA and H2A.Z co-localize at thousands of sites, where they interact both cooperatively and antagonistically to regulate gene transcription and the associated nucleosome dynamics [25].

### 2.2. ISWI Subfamily

In eukaryotes, ISWI proteins form conserved protein complexes with DDT (DNA-binding homeobox and different transcription factor) domain proteins. The resulting protein complexes can alter nucleosome positions without affecting the nucleosome structure (i.e., sliding the nucleosomes along the DNA strand) [26].The remodeling due to nucleosome-sliding requires the HAND, SANT, and SLIDE domains in the C-terminus of yeast ISWI proteins to interact with the linker DNA between nucleosomes, and may also involve the interaction between the SANT domain and the histone H3 tail. These interactions anchor ISWI proteins to nucleosomes and facilitate the movement of DNA along the surface of nucleosomes [27]. In metazoans, ISWI proteins can form many types of complexes with distinct DDT domain proteins, each playing a unique role in transcription, chromatin assembly, and the maintenance of higher-order chromatin structures [26,28,29]. The Arabidopsis genome encodes two ISWI proteins, CHR11 and CHR17, and at least 12 DDT domain proteins, many of which form complexes with ISWI proteins [30]. Micrococcal nuclease digestion coupled with high-throughput sequencing data revealed that the ISWI remodeling factors CHR11 and CHR17 in Arabidopsis help produce the evenly spaced nucleosome pattern in gene bodies [4]. Mutations to the *CHR11* and *CHR17* genes, as well as a deficiency in the functional DDT domain proteins RLT1/2 (RINGLET1/2), result in impaired vegetative organ development and early flowering [31].

### 2.3. CHD Subfamily

The chromodomain is a unique protein domain containing conserved hydrophobic amino acids that interact with methylated lysine residues. The CHD subfamily chromatin remodeling factors have an additional double chromodomain before the Snf2 domain [6]. Plants encode two types of CHD chromatin remodelers, CHD1 and CHD3. Both Arabidopsis and rice genomes encode one CHD1 homolog and three CHD3 homologs. The Arabidopsis CHD3 homologs are PKL (PICKLE) and PKR1/2 (PKL-related 1/2) [32]. Studies of functionally deficient mutants have highlighted that CHD1 can mediate post-replication chromatin assembly and nucleosome spacing, both independently and synergistically [33]. In addition to the double chromodomain, CHD3 also contains a methyl-histone interaction domain, namely PHD [34]. The PHD domain in many proteins specifically interacts with trimethylated H3K4. Similar to chromodomain interactions, this interaction involves two or four aromatic residues in the structural domain. Unlike SWI/SNF and ISWI proteins, Arabidopsis PKL exists mainly as a monomer [35].

### 2.4. INO80 Subfamily

Compared with other chromatin remodeling factors, the INO80 subfamily proteins have a considerably longer “insertion region” separating the N-terminal SNF2_N and C-terminal Helicase_C. This subfamily includes the chromatin remodelers INO80 and SWR1 (SWI2/SNF2-related 1), both of which are highly conserved in eukaryotes and function as integrated scaffolds that interact with many other subunits to form protein complexes (i.e., INO80-C and SWR1-C). Moreover, INO80-C shares many important and indispensable subunits with SWR1-C, such as RVB1 and RVB2, but there are also many complex-specific subunits, including ARP5 in INO80-C and ARP6 in SWR1-C [36]. In eukaryotes, SWR1-C functions specifically in replacing the H2A-H2B dimer within the nucleosome with H2A.Z-H2B, thereby depositing the histone variant H2A.Z into the chromatin. The recently resolved molecular architecture of yeast SWR1-C revealed a large protein complex containing 13 subunits, including SWR1 [37]. The composition of the corresponding plant SWR1-C has been determined, and the interplay of subunits in the protein complex has also been fully characterized at the genetic and biochemical levels [36]. Additionally, ARP6 is considered to be a highly specific subunit of SWR1-C. Recently, ARP6 was confirmed to interact with MBD9 (methyl-CpG-binding domain 9), which also strongly interacts with ISWI chromatin complexes [38]. Together with the crosstalk of BRAHMA and H2A.Z [25], these findings imply complex crosstalk involving distinct chromatin remodelers.

Unlike SWR1-C, INO80-C in eukaryotic organisms other than plants reportedly have diverse biological activities related to nucleosome eviction, establishment of nucleosome spacing, and removal of H2A.Z [39,40]. The Arabidopsis INO80 homolog AtINO80 interacts with ARP5 and forms larger protein complexes with other proteins. AtINO80 and ARP5 play a synergistic role in the cell proliferation associated with plant growth and development as well as in the maintenance of chromatin stability under replicative stress [41]. The recruitment of AtINO80 to local chromatin can be affected by the ARP6-mediated H2A.Z distribution. Genetic analyses of Arabidopsis INO80-C and SWR1-C indicated that they have very complex functional interactions and critical physiological roles influencing plant embryonic development [41]. Additionally, SWR1 and INO80 remodelers contribute to the DNA DSB (double-strand breakage) perinuclear anchorage site selection and play distinct roles in the DSB repair of budding yeast cells [42,43]. They also regulate the frequency of the DSB-repair pathway homologous recombination in Arabidopsis [44,45].

## 3. Plant Histone Chaperones

Because of the strong electrostatic interaction between DNA and histones at physiological ionic strength, the orderly formation of nucleosome structures requires histone chaperones. These chaperones block non-specific interactions between positively charged histones and negatively charged DNA and are important for nucleosome assembly/disassembly, which are very dynamic processes in fundamental biological activities such as transcription and DNA replication, repair, and recombination [46].

In eukaryotes, most histone chaperones are conserved regarding sequence and activity, and can be classified as H3-H4 or H2A-H2B chaperones. These two chaperone groups play specific roles in different nucleosome assembly/disassembly steps [47,48]. Histones are assembled into a nucleosome structure via a step-by-step process, starting with the deposition of a histone (H3-H4)_2_ tetramer on the naked DNA by the corresponding chaperones, followed by the H2A-H2B histone chaperone-mediated addition of the H2A-H2B dimer on each side. This ordered process is reversed during nucleosome disassembly. Notably, H3-H4, but not H2A-H2B histones, can be deposited directly on naked DNA, which represents a critical control point for the formation of ordered structures in nucleosomes. Recent studies on nucleosome asymmetry and sub-nucleosomal structures, such as hemisomes/half-nucleosomes (one copy of H2A/H2B/H3/H4) and hexasomes (two copies of H3-H4 and one copy of H2A-H2B), indicate that multiple pathways may be responsible for the subtle regulation of nucleosome structures. Moreover, these sub-nucleosomal structures may be important for transcription [49,50].

In plants, the most thoroughly studied histone chaperones with functions in the root apical meristem (RAM) are the CAF-1 (chromatin assembly factor-1) H3-H4 histone chaperone and the NRP (nucleosome assembly protein 1-related protein) H2A-H2B histone chaperones. Both were observed to assemble nucleosomes in vitro in earlier biochemical studies [47,48,51].

The H3-H4 histone chaperone CAF-1 was originally purified from the nuclear extract of human cells and was identified as a protein complex consisting of three subunits, p150, p60, and p48. The p150 subunit of human CAF-1 interacts directly with H3-H4 through its acidic region. This subunit also binds to PCNA (proliferating cell nuclear antigen), which recruits CAF-1 to replication- and DNA repair-coupled chromatin assembly sites [52,53]. The CAF-1 complex is relatively conserved in eukaryotes, and its plant orthologs are FAS1 (FASCIATA 1), FAS2, and MSI1 (multicopy suppressor of IRA1) [54]. The *fas1* and *fas2* mutants are recessive mutants that were isolated by forward genetics during the screening for mutations that lead to abnormal meristematic structures. Almost all of the organs of *fas1* and *fas2* mutant plants are affected, and the most obvious phenotype is the fasciation of plant shoots. The *fas1* and *fas2* mutants also exhibit decreased root growth [55]. Both FAS1 and FAS2 are unique subunits of plant CAF-1, and their production is highly associated with rapidly dividing tissues such as meristems. The transcription of CAF-1 subunit-encoding genes is regulated in the cell cycle, with peak levels in the S phase [54,56].

The NAP1 (nucleosome assembly protein 1) protein was first identified and purified from *Xenopus laevis* eggs as a factor promoting nucleosome assembly in vitro [57]. The NAP1 homologs were subsequently identified in various organisms, ranging from yeast to humans [58]. The NAP1 family proteins have a conserved central structure, non-conserved N-terminal extensions of varying lengths, and a highly acidic C-terminal tail. The central structure is necessary and sufficient for histone-binding and nucleosome assembly, as well as dimerization. The NAP1 family consists of two subfamilies, NAP1 and NRP, whose members differ regarding sequences. The NAP1 and NRP subfamily members can form homologous or heterologous dimers with other members of the same subfamily, but cannot form NAP1-NRP dimers. They also vary in terms of subcellular localization, with NRP proteins mainly located in the nucleus, whereas some NAP1 proteins are localized in the cytoplasm and others are shuttled between the nucleus and cytoplasm. In Arabidopsis, both types of histone chaperones can specifically bind H2A-H2B and help regulate the frequency of somatic cell homologous recombination and transcription [51].

## 4. RAM (Root Apical Meristem)

The cell fate determination and pattern formation in meristems during organ development have always been key issues in studies of multicellular development. The root tip of the model plant Arabidopsis is an excellent research model because of its simple meristem organization, as well as the fact there are thoroughly analyzed Arabidopsis cell lineages with considerable genetic information. The SCN (stem cell niche) in the RAM is produced by the initial cells surrounding a number of relatively stationary stem cells that form the QC (quiescent center) [59]. The surrounding initial cells are mitotically active, including the CSC (columella initial/stem cell), Epi/LRC (epidermal cell and lateral root cap) initials, CEI (cortical/endodermis initial), and stele initials. After dividing, the initial cells regenerate through asymmetric division and produce daughter cells that migrate from the SCN [60]. The asymmetric division of the distal CSCs at the root tip directly generates differentiated columella cells containing gravity-sensing starch granules to replenish the continuous loss of root cap cells. The Epi/LRC initial cells divide first anticlinally and then periclinally, after which they differentiate into the epidermis and LRC, respectively. The CEIs divide periclinally and differentiate into two types of ground tissues, namely the cortex and endodermis. The amplification of stele initials results in the production of a vascular cylinder [59]. These proximal initial cells rapidly divide in the meristem regions, and their progenies decrease their rate of division and begin to undergo endoreduplication. Meanwhile, they apparently elongate in the elongation zone and eventually differentiate in the differentiation zone, forming the morphological features of a radial root structure at maturity [61]. Similarly, rice RAM is composed of three histogens, calyptrogen (peripheral root cap and columella initial cells), dermatogen-periblem complex (epidermis-endodermis initial cells), and plerome (stele initial cells), surrounding a putative QC [62]. Root tips consist of various cells with distinct cell proliferation rates and cell fates. There is increasing genetic evidence that chromatin remodeling factors and histone chaperones affect replication-coupled chromatin stability, transcriptional regulation, and cell fate determination in root apical meristems [12].

## 5. Implications of Chromatin Remodeling Activities in the RAM

High-throughput experiments combined with bioinformatics analyses have established the transcriptome patterns in the RAM and generated some information regarding the transcriptional regulation in the RAM, thereby considerably increasing our understanding of the framework of the complete gene regulatory network during RAM development [63]. A previous study proved that WOX5 (WUSCHEL-RELATED HOMEOBOX-5) is a WUSCHEL-related homeobox transcription factor that is produced only in the QC [64]. Additionally, WOX5 is the key regulator of the QC identity and promotes the local accumulation of the phytohormone auxin. The distribution of auxin in the RAM is partly the result of the polar auxin transport mechanism via the auxin efflux carrier PIN (PIN-FORMED). Moreover, PLT (PLETHORA) proteins are AP2 domain-containing transcription factors, among which PLT1 and PLT2 are important for root development [65]. The PLTs are produced throughout the RAM, forming a maximum gradient near the root tip. The formation of such a gradient is reportedly a prerequisite for root zonation in Arabidopsis. High *PLT* expression levels promote stem cell maintenance, whereas intermediate expression levels enhance the mitotic activity in the RAM. Outside of the elongation zone, low *PLT* expression levels are required for cell elongation and differentiation [66]. The incorrect specification of the QC and the dysfunction of initial cells were observed in *wox5* and *plt* mutants, implying these genes are important for determining the QC cell fate and the maintenance of the SCN.

The SWI/SNF remodeling factor *brm* mutants have a defective root SCN and exhibit decreased meristematic activity and root growth retardation. Moreover, BRM maintains the root SCN by directly targeting the chromatin of several *PIN* genes (*PIN1*–*PIN4* and *PIN7*) and altering their expression. Furthermore, the overexpression of *PLT2* can partially rescue the root SCN defects in *brm* mutants [67]. Notably, BRM also regulates the balance between primary root growth and stress responses through the abscisic acid (ABA) signaling pathway. BRM can bind to the chromatin region of the ABA-induced basic domain/leucine zipper transcription factor gene *ABA INSENSITIVE 5* (*ABI5*), and the anchored BRM helps to stabilize a nucleosome that likely represses *ABI5* transcription under non-inductive conditions. The *brm* mutants exhibit ABA sensitivity and increased drought tolerance, similar to the phenotype of *ABI5*-overexpressing transgenic plants [68].

The SWI/SNF complex SWP73-type subunit BAF60 can repress the transcription of *IPT3/IPT7* (*ISOPENTENYLTRANSFERASE 3* and *7*), the key genes in cytokinin biosynthesis, and the CDK (cyclin-dependent kinase) inhibitor gene *KRP7* (*KIP-related protein 7*) by inhibiting the deposition of the active histone mark H3K4me3 and chromatin loop formation, thereby controlling cytokinin production and cell cycle progression to maintain the RAM and promote root growth [69]. Nevertheless, the molecular mechanism linking BAF60 activity with histone modifications remains to be elucidated.

During plant development, *MINU1/CHR12* and *MINU2/CHR23* play redundant roles related to RAM establishment and maintenance. The phenotypes of the *chr12* and *chr23* single mutants are similar to the wild-type phenotype. In contrast, the *chr12 chr23* double mutant, in which both genes are knocked out, exhibits plant embryo lethality and fails to produce embryonic meristems. Although a weak double mutant can germinate, the observed abnormal growth direction and the impaired rate of cell division result in a significantly defective RAM structure with an unrecognizable QC and an irregular columella cell layer [70]. Additionally, MINU2/CHR23 can bind directly to the *WOX5* promoter and repress expression. In contrast, the overexpression of *CHR23* inhibits root growth because of decreased cell elongation [71].

Chromatin remodeling and histone modifications are involved in many aspects of root development. In animals, CHD3 family members are included in the NuRD (nucleosomal remodeling and deacetylase) complex, which helps repress transcription by coupling chromatin remodeling and deacetylation activities [72]. Most studies on plant CHD3s have involved the model species Arabidopsis and rice. In Arabidopsis, the CHD3 chromatin remodeling factor PKL is crucial for root development. It inhibits the expression of seed maturation genes and prevents the development of embryonic traits in postembryonic roots [73]. Moreover, PKL and histone acetylation contribute to the inhibition of auxin-mediated lateral root initiation [74].

An earlier investigation indicated PKL is necessary for establishing and maintaining the root meristem, with *pkl* mutants producing short roots with low root meristem activity [75]. Mutations to *PKL* result in the low penetrance of the “pickle root” phenotype, characterized by embryonic-like primary roots and the abnormal accumulation of seed storage reserves in the roots [73,76]. These phenotypes are similar to those of mutants with an abnormal histone H3K27 methylation-related PcG (polycomb group) [77]. Chromatin immunoprecipitation experiments proved that PKL can directly target the PcG genes *SWN* and *EMF2* in roots. The deletion of *PKL* results in decreased PcG-mediated H3K27me3 levels and the increased expression of many PcG target genes in the roots, similar to the effects of mutations to PcG [76]. Independent studies have suggested that PKL promotes H3K27me3 and PcG functions [78,79]. However, PKL can counteract the PcG protein CLF (CURLY LEAF)-mediated transcriptional inhibition in root stem cells [75], indicating a complex interplay between PKL and other chromatin modifiers during RAM establishment and maintenance.

In rice, the PKL orthologous gene *CHR729* regulates root growth through the GA (gibberellin) pathway. The bioactive GA_3_ content is reportedly decreased in the *chr729* mutant, but the phenotype of this mutant may be partially recovered by the application of exogenous GA_3_ [80]. Moreover, CHR729 can interact with H3K4me2 and H3K27me3 through its chromodomain and PHD domains, respectively. A loss-of-function mutation to *CHR729* results in an overall decline in H3K27me3 and H3K4me3 levels, but not H3K4me2 and H3K4me1 levels [81].

In Arabidopsis, double mutations to *CHR11* and *CHR17* or to the DDT domain genes *RLT1/2* lead to the production of short roots [82]; however, the underlying mechanism remains unknown. Floral MADS-domain transcription factors can form a stable protein complex with CHR11/17 in the presence of DNA during the development of Arabidopsis flowers [83], which represents a possible mechanism for the effects of the ISWI protein on RAM maintenance. Alternatively, it may be the simple consequence of a disturbed genome-wide nucleosome distribution and cell fate misspecification in RAM stem cells [4]. A previous study revealed that AtINO80 acts in concert with AtARP5 during the cellular proliferation in developing roots [41]. Although the S-phase-specific association of INO80 with the initiation of replication has been demonstrated in yeast [84], a similar mechanism does not appear to be conserved in plants. Thus, future investigations should examine the potential link between chromatin remodeling activities and the progression of replication.

## 6. Roles of Histone Chaperones Related to RAM Development

As in mammals, H3 histones in plants are classified as the canonical histone H3.1 and the histone variant H3.3. Although these histones share similar sequences [85,86], their effects on chromatin structures are quite different. Histone variant H3.3 can be incorporated into chromatin outside the S phase and its function is generally related to transcriptional activation, whereas canonical histone H3.1 is only expressed in the S phase and is incorporated in chromatin in a replication-coupling manner mediated by the CAF-1 complex [87].

Mutations to *FAS1* and *FAS2*, which encode components of the Arabidopsis CAF-1 complex, result in inhibited root growth because of a decrease in mitotic activity as well as an abnormal SCN structure [54,56,88]. The incorporation of CAF-1-mediated H3.1 in the S phase ensures the re-construction of daughter chromatin after replication, protects the genome from unexpected DNA damage (e.g., breakage), and maintains genomic integrity and stability. Accordingly, Arabidopsis *fas1* and *fas2* mutants exhibit chromosomal abnormalities during mitosis, such as the production of poly-centromeric chromosomes at metaphase as well as chromatid bridges and acentric fragments during anaphase-telophase [89]. A small, but significant, increase in the extent of DNA DSB was reported for *fas* mutants [90]. In these mutants, the expression levels of genes induced by DSB, such as DSB repair genes, were also significantly increased [56,88,91], further demonstrating that the CAF-1 activity associated with DNA replication helps control genomic stability in RAM stem cells.

Stem cells generate two daughter cells with different identities, allowing them to simultaneously self-renew and produce differentiated daughter cells. The influence of chromatin dynamics in this process deserves a detailed investigation. In the RAM of *fas1* and *fas2* mutants, the identities of the initial cells are reportedly relatively undefined, and many initial cells exhibit diverse characteristics [54,92]. Similarly, in animals, the differential inheritance of chromatin status is considered to be the basis of the asymmetric division of GSCs (germline stem cells), leading to diverse cell identities. Knocking out the large subunit of CAF-1, p180, causes the GSCs to retain stem cell characteristics, but express various markers [93], demonstrating that dynamic chromatin changes mediated by CAF-1 play a conserved role in regulating stem cell identities and genomic integrity.

Double-strand breakage can promote the early initiation of endoreduplication by inducing the cell cycle G2/M arrest [94]. The smaller root meristems of *fas* mutants also imply an early shift from mitotic activity to endoreduplication [54,56]. Abnormal endoreduplication due to impaired genomic stability was also observed in cell elongation and differentiation zones of *fas* mutants, resulting in a higher ploidy level of chromosomes, which is exemplified by enlarged epidermal cells and increased trichome branching on leaves [95].

The NRP1/2 histone chaperones are specific to H2A-H2B, with both ubiquitously expressed in plant organs. Single mutations to *NRP1* or *NRP2* do not result in obvious changes in development, but mutants in which both genes are mutated exhibit a short-root phenotype during postembryonic development, with a cell cycle arrest at G2/M in the root tips as well as disordered cell organization in the SCN. The double mutant is highly sensitive to genotoxic stress, with elevated levels of DNA damage and a lack of transcriptional gene silencing, implying that NRP1/2 contribute to chromatin stability and meristem maintenance in developing roots [96]. Recently, NRP1 was confirmed to interact with the transcription factor WEREWOLF (WER) in vivo and accumulate at the promoter regions of *GLABRA2* (*GL2*), which is the WER downstream target gene that is critical for epidermal cell specification. The WER-dependent enrichment of NRP1 at the target promoter is accompanied by histone eviction and nucleosome loss, which is an example of a molecular mechanism mediating the recruitment of a histone chaperone by a gene-specific transcription factor to modulate the chromatin structure for proper cell differentiation [97].

The H2A-H2B histone chaperone NRP1/2 and the H3–H4 histone chaperone CAF-1 play a synergistic role in SCN maintenance. The two chromatin factors coordinately regulate transcription. In the *nrp1/2 fas2-4* double mutant, in which the middle subunit of CAF-1 is mutated, more than 8% of the Arabidopsis genes are incorrectly transcribed, including several genes related to root meristems, such as *WOX5* and *SHR* (*SHORT ROOT*), and auxin-related *ARF5* (*AUXIN RESPONSE FACTOR 5*) and *IAA30* (*INDOLE-3-ACETIC ACID INDUCIBLE 30*). In the roots of the *nrp1/2 fas2-4* triple mutant, the expression level of the auxin-response reporter gene fused to the green fluorescent protein (GFP) gene (*DR5rev:GFP*) was significantly downregulated, and the pattern of peak distribution at the QC was also largely lost. Thus, an abnormal auxin regulation pathway is likely responsible for the defective root development of the *nrp1/2 fas2-4* mutant. Moreover, NRP1/2 and CAF-1 are considered to coordinately function to control chromatin replication and genomic integrity and prevent cell death in root meristems, which is essential for normal SCN functions [92].

A similar mechanism can also be found in the genetic interaction between *NRP1/2* and the chromatin remodeling factor *INO80*. The *nrp1/2 ino80* double mutant exhibits an extreme short-root phenotype and hypersensitivity to the auxin transport inhibitor NPA. These two chromatin factors are recruited to the chromatin regions of the auxin efflux carrier gene *PIN1*, and mediate the local histone occupancy, thereby modulating the transcription level. Their combined functions are also implicated in chromatin replication and the maintenance of genomic integrity to ensure the viability of root meristem cells [98]. However, the functional interplay between plant INO80 and CAF-1 has not been analyzed yet.

## 7. Perspective

Currently, the studies on chromatin remodeling factors and histone chaperones in the RAM are mainly focused on replication-coupled chromatin stability and the transcriptional regulation associated with cell fate determination. However, because of the substantial abundance of chromatin remodelers and histone chaperones and their potential functional redundancies, many well-known factors are not associated with an obvious phenotype in the RAM. Future genetic analyses and investigations involving single-cell sequencing technology will help clarify the epigenetic pathways related to the initial cell specification and proliferation in the SCN and their differentiation. Moreover, other important issues still need to be resolved. For example, how chromatin factors specifically target genes and how they form complex regulatory networks will need to be determined. Future investigations of the chromatin remodeler–histone chaperone interactome and comprehensive chromatin immunoprecipitation sequencing analyses will be useful for elucidating the mechanism by which plant root tips precisely respond to environmental stimuli and developmental needs. Furthermore, the available techniques for in situ chromatin studies are critical for resolving the nuclear architecture [99]. High resolution microscopy enables the three-dimensional examination of the plant nuclear architecture and the characterization of the spatial dynamics of the nuclear compartment, specifically regarding the chromatin remodeling factors and histone chaperones.

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
