# Peer review of "Activities of Chromatin Remodeling Factors and Histone Chaperones and Their Effects in Root Apical Meristem Development"

_ijms, 2020, doi:10.3390/ijms21030771_

Round 1
Reviewer 1 Report
The manuscript provides an easy-to-read review of the chromatin remodeling factors and histone chaperones related to root apical meristem. The text is well organized, and the flow of information is logical, and with sufficient literature. Actually, I would include some more figures and schemes, that are generally welcome in such types of review, especially for scholars not familiar with the topic and thus desirous to learn. English is very good. Therefore, I recommend accepting the manuscript.
Author Response
Response: We thank Reviewer 1 for the encouragement.
Reviewer 2 Report
We are now move to epigenetic era that’s why this review is important update for our knowledge in the topic.
This is potentially interesting review, howver, many corrections still required before accetance. Below are some points:
Line 16: Please, re-formulate more clearly: “differentiation of stem cells in meristems”
Line 25: “In plant growth and development, precise modulation of chromatin structure in the nuclei is necessary to coordinate with the transcriptional reprogramming in response to external stress or internal signaling pathways, especially in cell proliferation and cell fate determination within meristem regions.”
I would suggest to not mixing stress and normal development. Chromatin structure is a driver of cell specification in the RAM.
Moreover, so many reviews cited by authors were described chromatin structure in animal cells. This is OK, but authors have to mentioned these facts and in the first part give comparison between two kingdoms. Anyway, authors have to mention about each cells they are talking in each cases.
Line 53: Is it better to avoid authors name in the text or insert in all citations.
Line 61: One can not “summarize the subfamilies“, please, edit.
Line 65 It is better to write: “The genome of model plant Arabidopsis”
Line 159; “They play a synergistic role in cell proliferation in the context of plant growth and development, and in the maintenance of chromatin stability under replicative stress.“- can you please, provide citations to these statements?
Line 190: “Both are named after their activities in nucleosome assembly in vitro in biochemical studies” – please, edit.
Line 201: “highly correlated with meristem and rapidly dividing tissues”- please, edit..
Line 202: Genes can not regulated by cell cycle, its level can only correlate with certain cell cycle stages. “Genes encoding the CAF-1 subunit are also shown to be regulated by cell cycle with their peaks in S phase [54, 56].”
Line 213: “cytoplasmic distribution“ – it is better to write localization.
Line 217: the description of root apical meristem in not clear. Authors did not described proximal and distal meristem and chromatin organization in each part. Root zonation also not well described, including differences in chromatin structure between inner and outer cell layers.
Line 237; what is “after RAM area”?
Line 244: Well-developed in situ method allow to study chromatin structure in situ with cellular resolution in the frame of organ coordinate system.
Figure 1: does not provide any useful information. You can either re-organize with more information, or remove it.
Line 267: “This gradient is reported to be a prerequisite for root zonation” – gradient can not be a prerequisite.
Line 270: “After elongation zone, cell elongation and differentiation require PLT expression levels under a certain threshold”- please, edit.
Line 331: “Roles of histone chaperones in RAM” development.
Line 400: In perspective part it will be nice to include the modern technique for in situ chromatin study.
Author Response
Response: We feel sorry that we did not make it easy for readers to read the manuscript, and we thank Reviewer 2 for so many helpful advices on the writing. We have removed the figure, included the in situ technique in perspective, and edited all the texts according to the suggestions in the revised manuscript. We hope that our efforts has effectively improved the quality of this manuscript.
Reviewer 3 Report
I found this article to be interesting and informative. I did note numerous minor grammatical errors and some formatting errors (i.e. failure to capitalize of scientific names, failure to define some abbreviations/acronyms, etc.). However, these are easily corrected.
Author Response
Response: We thank Reviewer 3 for the encouragement and suggestions. We have edited the texts in the revised manuscript, and we hope that our efforts has effectively improved the quality of this manuscript.
Round 2
Reviewer 2 Report
The text is somehow imporved, but more clarity still require,
Title: it is better to write “root apical meristem development.”
Line 19: “epigenetic regulation in root apical meristem development”
Line 21: mention rice, but there are no informations about rice RAM in the text.
Please, clarify species you describe in each sentence, For example, citation 9 nothing to do with plants, but title of the this part “Plant chromatin remodeling factors”.
Citation 16 is also not plants. It is better to make order in these descriptions. May be at the forst stage give examples from non-plants and thereafter describe plants part.
Lines 214-236: if possible, please, improve description of the RAM of Arabidopsis.
Line 423: “Nucleic Acids Res 2006, 34, (10), 2887-905“ ??
Author Response
Response: We thank Reviewer 2 for so many helpful advices on the writing. According to the suggestions, we have edited the corresponding texts, included new citation and needed clarification, and corrected the mistakes in references in the revised manuscript. We hope that our efforts has effectively improved the quality of this manuscript.